
# Detecting plumes in mobile air quality monitoring time series with Density-based Spatial Clustering of Applications with Noise

Blake Actkinson[1], Robert J. Griffin[1,2*]

[1]Department of Civil and Environmental Engineering, Rice University, Houston, TX 77005, USA

[2] School of Engineering, Computing, and Construction Management, Roger Williams University, Bristol, RI 02809, USA

*Correspondence to*: Robert Griffin (rgriffin@rwu.edu)

**Abstract.** Mobile monitoring is becoming an increasingly popular technique to assess air pollution on fine spatial scales, but methods to determine specific source contributions to measured pollutants are sorely needed. One approach is to isolate plumes from mobile monitoring time series and analyze them separately, but methods that are suitable for large mobile

monitoring time series are lacking. Here we discuss a novel method used to detect and isolate plumes from an extensive mobile monitoring data set. The new method relies on Density-based Spatial Clustering of Applications with Noise (DBSCAN), an unsupervised machine learning technique. The new method systematically runs DBSCAN on mobile monitoring time series by day and identifies a subset of points as anomalies for further analysis. When applied to a mobile monitoring data set collected in Houston, Texas, analyzed anomalies reveal patterns associated with different types of

vehicle emission profiles. We observe spatial differences in these patterns and reveal striking disparities by census tract. These results can be used to inform stakeholders of spatial variations in emission profiles not obvious using data from stationary monitors alone.

*Graphical Abstract*

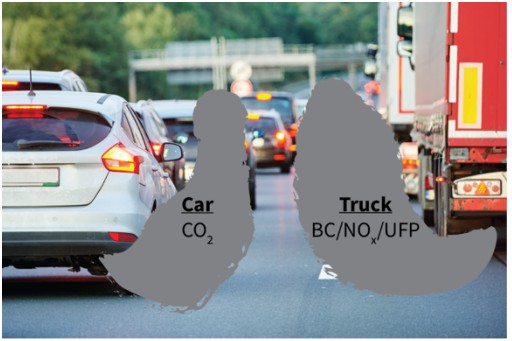

## 1 Introduction

A central question of air pollution studies is to identify the varied sources that contribute to measured pollutant concentrations. This question becomes more complicated in a mobile monitoring context because measurements and


concentrations vary as a function of both space and time, making conventional source apportionment techniques such as positive matrix factorization and principal component analysis (PCA) harder to apply effectively (Larson et al., 2017).

Recently published work took several approaches to performing source apportionment on measured pollutants in a mobile monitoring context. One approach involves using PCA on background subtracted measurements, such as in Larson et al. (2017), whose approach has limitations when applied to extensive mobile monitoring campaigns because it defines a rolling minimum across a static time window that may not be applicable for extensive mobile monitoring campaigns with ≈ 20-30x the temporal coverage. Other approaches have focused on using Land Use Regression (LUR) models to identify

relationships between pollutants and land use variables, such as in Messier et al. (2018). However, LUR models require spatiotemporal databases of sufficient temporal and spatial resolution for use in model training. While recent efforts have illustrated creative methods of creating these land use databases (Qi and Hankey, 2021), use of these models is still limited through the availability of these databases. There is a need for the development of methods that can identify source influences in large mobile monitoring data sets at high time resolution without being subject to the availability of land-use

variable databases.

Another factor that aggravates source identification in mobile monitoring contexts is the nature of mobile monitoring data themselves. If a mobile monitoring campaign were conducted focusing largely on residential areas with brief excursions into traffic congested areas, such as highways, performing PCA or other dimension reduction techniques to describe patterns in the entire dataset would likely return results that are weighted towards residential areas with negligible source influences.

This type of analysis generates solutions in which there is a demarcation between a majority of points with little source influence and a smaller subset of source-influenced points elevated in all pollutants, which is not compelling if one's objective is to determine the specific sources affecting the measurements.

This raises the question of how to identify source influences within mobile monitoring time series that cover locations ranging from 'background' to 'highly influenced by sources.' If one could identify source spikes or plumes within mobile

monitoring time series, one could restrict their analysis to these plumes to categorize the different types of sources that affected their mobile monitoring measurements. Plume identification within mobile monitoring time series has been addressed previously. Hagler et al. (2012) use a rolling coefficient of variation across a 5-s time interval, then flag points with a coefficient greater than 2. Drewnick et al. (2012) use a different moving window algorithm that calculates the standard deviation of points below a defined background threshold ($\sigma_b$) and flags points which are more than $3\sigma_b$ above the

previous point. The algorithm then flags subsequent points, increasing the threshold necessary (by a factor of $\sqrt{nf}$, in which $nf$ is the total number of flagged points) for flagging for every subsequent point beyond the first flagged. Others have addressed the plume identification question indirectly through background estimation and removal methods.

These methods all have drawbacks. In the data used in the present work, the method of Hagler et al. (2012) flags few to no points at all, suggesting that the method is sensitive to the time series utilized. The algorithm of Drewnick et al. (2012)

suffers in situations where many plumes appear consecutively to one another, frequently leading to poor performance in



those circumstances. Other methods depend on a time window, which presents problems for complex, multi-day mobile monitoring time series.

Here we discuss an algorithm to identify plumes in a different manner. The algorithm relies on Density-based Spatial Clustering of Applications with Noise (DBSCAN), a nearest neighbor clustering algorithm (Ester et al., 1996). DBSCAN

clusters points based on whether they fall into predetermined neighborhoods with other points. The technique can cluster points with more complicated shapes (e.g., an "S" embedded in noise in two-dimensional space) and is not sensitive to starting values compared to other clustering techniques such as k-means (Tan et al., 2019). Additionally, the algorithm does not require every single point to be clustered, allowing for those points that do not neatly fall into a given cluster to be defined as noise.

The objective of this work is to establish a new method for detecting plumes in mobile monitoring time series, validate its performance, and use it to perform novel analysis that elucidates the impacts of different emission sources across census tracts in the Greater Houston area. We utilize DBSCAN by envisioning daily mobile monitoring time series collected in Houston (Miller et al., 2020; Actkinson et al., 2021) that include black carbon (BC), carbon dioxide ($CO_2$), oxides of nitrogen (nitric oxide (NO) + nitrogen dioxide ($NO_2$) = $NO_x$) and ultrafine particle number concentrations (UFP) as large

numbers of points clustered around a four-dimensional origin with plumes scattered outwards from this origin. In the DBSCAN context, plumes would be labeled as noise. We first describe DBSCAN, then detail how we adapt it for application to mobile monitoring time series. To evaluate performance, we construct a validation set by manually flagging plumes via visual inspection from a randomly chosen subset of days from the Houston mobile monitoring campaign (Miller et al., 2020; Actkinson et al., 2021). We use the validation set to tune DBSCAN and other time series-based models and

compare performance of all models. We apply the algorithm to the Houston mobile monitoring dataset to identify anomalies, which are then clustered into anomaly types linked to specific vehicle emission sources. We tabulate the number of these different anomaly types by census tract and derive anomaly frequencies, which are conceptualized as the probability of detecting a given anomaly type during the prescribed study period. We demonstrate differences in anomaly frequencies in census tracts across Houston, which can be used to tailor census-tract specific air monitoring regulation and enforcement

strategies. We discuss the implications of the method, the results, and future directions for this research.

## 2 Methods

### 2.1 Data

Data were collected during the Houston mobile monitoring campaign and are described in detail elsewhere (Miller et al., 2020; Actkinson et al., 2021). The campaign's objective was to measure air pollution on a very fine spatial scale in 35

different census tracts across the Greater Houston area in a 9-month timespan. Two Google Street View cars were driven through these census tracts systematically to evaluate spatial differences in the concentrations of 7 pollutants. Previous analyses with this dataset focused on identifying large concentrations attributable to sources along specific individual



roadways and on developing a technique to identify and remove background concentrations from the time series collected (Miller et al., 2020; Actkinson et al., 2021).

In the current analysis, we restrict the set of analyzed pollutants to be BC, $CO_2$, UFP, and $NO_x$. Here, we do not consider fine particle mass ($PM_{2.5}$) concentration and ozone due to the influence of secondary processes. Table S1 in the Supplemental Information provides the instruments used to measure each respective pollutant. BC, $CO_2$, and UFP measurements were taken on 1-s time resolution, while NO and $NO_2$ measurements were taken on 5-s time resolution. With the addition of logged global positioning system (GPS) coordinates from each car, the campaign generated a massive spatiotemporal dataset

spanning millions of observations across the 9-month span.

In this work, we create a multivariate dataset consisting of the four air pollution variables at 1-s time resolution, along with corresponding latitude/longitude coordinates and timestamps that span 277 separate days of sampling for a total of 5,301,507 observations. The BC data were smoothed with a 10-s time window to limit the effects of noise on subsequent analysis. In the original data set, NO and $NO_2$ were taken on a 5-s time resolution, while $CO_2$, BC, and UFP were all collected at 1-s

resolution. To perform analysis at a finer temporal resolution, as well as to address missing data, we use monotone Hermitian splines to impute missing measurements up to a 6-s time gap. While previous mobile monitoring studies have fused 5-s data with 1-s data by repeating the same 5-s measurement each second across the entire interval (Shah et al., 2018; Miller et al., 2020), we argue that using continuous splines provides a more realistic estimate of missing 1-s information in this context. Previous studies have focused on preserving the spatial meaning of concentration plotted on maps at very fine spatial

intervals; here, we are more interested in estimating temporal variations in missing concentrations, and splines are suitable tools to do so for brief, 6-s intervals.  Total imputed percentages for each pollutant were 1.06%, 80.0%, 80.0%, 0.42% and 0.49% for BC, NO, $NO_2$, $CO_2$, and UFP respectively; 90.1% of $NO_x$ realizations had at least one imputed measurement. Any multivariate realization with at least one missing observation in a variable not imputed was excluded otherwise. Days in which the cars operated had to possess a minimum of 600 measurements to be included in the analysis. Using road shapefiles

available through the TigerLINE road database (2020), we assign road categories to each of our points based on their respective latitude and longitude coordinates. To be consistent with Miller et al. (2020) and Actkinson et al. (2021), we restrict our analysis to points with logged latitude/longitude coordinates on primary, secondary, local, and private roads, as well as ramps and service drives because these are roads typically relevant to an individual's exposure. To account for GPS error, we remove logged GPS coordinates whose nearest neighbor distance to a TigerLINE shapefile point is more than 30

m.  Additionally, we observed evidence of the vehicles sampling their own exhaust when driving to and from dead ends in a previous analysis of the dataset (Miller et al., 2020). Because we do not want to characterize our own individual vehicle's emissions, we remove  points less than 30 m from a dead end in a road.

## 2.2 DBSCAN

DBSCAN is a clustering routine originally conceived by Ester et al. (1996). Using two predefined parameters, epsilon ($\epsilon$)

and *MinPts*, DBSCAN seeks to label points that have *MinPts* points within a neighborhood defined with radius $\epsilon$ as core





points, points that do not meet the *MinPts* criteria but have a core point within their $\epsilon$-neighborhood as border points, and points that do not fit either of these criteria as noise.

More formally, the $\epsilon$-neighborhood around a point $p \in D$ is defined using the notation of Hahsler et al. (2019) as

$$N_\epsilon(p) = \{q \in D | d(p,q) < \epsilon\} \tag{1}$$

where $N$ is the neighborhood, $D$ is the set of points, and $d$ is a distance measure such as the Euclidean distance. A point is defined as a core point if

$$|N_\epsilon(p)| \geq MinPts \tag{2}$$

where *MinPts* is the minimum points parameter and $\|$ denotes cardinality. The algorithm systematically labels points as core points, border points, or noise points depending on these criteria.

**2.3 Validation Set Construction**

To tune parameters and evaluate algorithm performance, we construct a validation set from the mobile monitoring data by manually flagging visible plumes within 30 randomly selected daily mobile monitoring time series (out of a possible total of 277); example validation set data are shown in Fig. S1. The total number of points in the validation set was 564,107, which amounts to $\approx 10\%$ of the entire set. A graphical user interface in IgorPro was used to flag plumes by visually inspecting the time series for spikes in pollutant concentrations for each pollutant (BC, $CO_2$, $NO_x$, and UFP). Any time series realization that had a spike in at least one pollutant was flagged.

**2.4 Algorithm Description**

We create an algorithm incorporating DBSCAN to label anomalies systematically within the Houston mobile monitoring campaign. Pseudocode for this algorithm is given in Fig. 1. The algorithm estimates $\epsilon$ and *MinPts* parameters for daily time series in the campaign based on the number of points in each time series and its dispersion and subsequently performs DBSCAN using these estimated parameters. We define the *MinPts* parameter to be the product of the total number of points in the daily time series, $n$, and a fractional value parameter, $f_{val}$. We set $f_{val}$ to 0.03 using the external validation set and describe the specific procedure in Sect. 2.6. We do not consider values of $f_{val}$ greater than 0.5 due to rapidly increasing computational cost and poor performance at higher values. After calculating *MinPts*, we determine $\epsilon$ using a k-nearest-neighbor (knn) distance ordering procedure in which the value of $k$ was set equal to *MinPts* and in which a point is the kth nearest neighbor to another point if the distance between the two points is the kth shortest distance among all points. We construct an ordered knn distance set and determine the mean and standard deviation of the first 30 ordered distances, then



define $\epsilon$ as the first distance that is greater than the mean plus 3 times the standard deviation of the subset of previously ordered distances. We iterate through the entire set of remaining distances, adding the current distance to the subset if it does not meet the criteria used to define $\epsilon$. Once both $\epsilon$ and *MinPts* are determined, we run DBSCAN on the daily time series

observations in which core points are labeled as normal and both border and noise points are labeled as anomalies. An example of labeled DBSCAN output for a scatterplot of daily $BC/CO_2$ time series is given in Fig. 2.

---

**DBSCAN Algorithm: Algorithm to identify and classify anomalies**

***Input:*** *Daily time series (for a given mobile platform if multiple)*
***Output:*** *DBSCAN-labeled anomalies conceptualized as plumes*
***Initialize*** *labeledAnoms \\ empty vector = number of total points in mobile TS*
***For*** *(each daily time series) \\ determine the parameters eps and MinPts and run DBSCAN*

| | |
|---|---|
| **1** | ***Scale*** *each variable to mean 0 and variance 1* |
| **2** | ***Set*** *minPts = 0.03 * n \\ n is the total number of points in the daily mobile monitoring time series* |
| **3** | ***Construct*** *knn ordered distance graph with k = minPts* |
| **4** | ***Set*** *dists = first 30 ordered distances* |
| **5** | ***Set*** *mean = mean of dists* |
| **6** | ***Set*** *sd = standard deviation of dists* |
| **7** | ***Set*** *d = $31^{st}$ distance in ordered set of distances* |
| **8** | ***For*** *(d, d < total number of distances, d++) \\ Go through remaining distances in the ordered set and find the first distance that is greater than the mean + 3 standard deviations of the set of previous ordered distances* |

   ***If*** *(d > mean + 3 * sd)*
      ***Set*** *eps = d*
      ***Break***
   ***Else*** *\\ Add d to the subset of dists*
      ***Concatenate*** *d to dists*
      ***Set*** *mean = mean of dists*
      ***Set*** *sd = standard deviation of dists*
***End***
*\\ With eps and MinPts, run DBSCAN on the daily time series*

| | |
|---|---|
| **9** | ***Set*** *dbOutput = dbscan(daily time series, minPts, eps) \\ dbOutput returns DBSCAN labeled core, border, and noise points* |
| **10** | ***Set*** *labeledAnoms = 1* ***if*** *dbOutput is core* ***else*** *2 if dbOutput is border, noise* |

***End***

---

**Figure 1. Pseudocode for the DBSCAN Plume detection algorithm.**






**Figure 2. Daily scatterplot example of DBSCAN labeled anomalies (red) for $CO_2$ against BC. Points labeled as normal (black and clustered near the origin) are ≈ 2/3 of the time series realizations in this example.**

### 2.5 Description of Other Algorithms

To put the performance of the DBSCAN anomaly detection algorithm in context, we compare its labeled anomalies with
output from the previously described plume detection technique of Drewnick et al. (2012) (referred to as "Drewnick"
moving forward) or base-case $90^{th}$-quantile algorithms. These two base-case algorithms, the Quantile-OR (QOR) and the
Quantile-AND (QAND) algorithms, flag points as anomalous based on criteria centered around the $90^{th}$ quantile of pollutant
distributions. In the QOR case, points are flagged as anomalous if any one pollutant measurement (BC, $CO_2$, $NO_x$, or UFP) is
above the $90^{th}$ quantile for the given daily time series (if $BC_t > 90^{th}$ BC OR $CO_{2,t} > 90^{th}$ $CO_2$ OR $NO_{x,t} > 90^{th}$ $NO_x$ or $UFP_t >$
$90^{th}$ UFP). In the QAND case, points are flagged as anomalous if *all* pollutant measurements are greater than their respective





90th quantiles (if $BC_t$ > 90th BC AND $CO_{2,t}$ > 90th $CO_2$ AND $NO_{x,t}$ > 90th $NO_x$ AND $UFP_t$ > 90th UFP). We run these algorithms, along with the Drewnick algorithm, on all daily time series to assess performance.

**2.6 Using the External Validation Set to Tune Parameters and Evaluate Performance**

To determine an appropriate value of $f_{val}$ for use in the DBSCAN algorithm, we perform grid search on values in [0.01, 0.10] in increments of 0.01 and [0.15, 0.50] in increments of 0.05. We do not consider values above 0.5 due to computational cost and poor performance at higher values of $f_{val}$. We evaluate performance using percentage agreement, defined as

$$\frac{\sum_i^N I(P_i = V_i)}{N} * 100 \qquad (3)$$

where $I(.)$ is the indicator function that evaluates to 1 if the condition is true and 0 otherwise, $P_i$ is the prediction label at point $i$, $V_i$ is the validation set label at point $i$, and $N$ is the total number of points in the validation set. Tuning results indicate that a value of 0.03 is most appropriate for $f_{val}$, which we use in subsequent analyses. In addition to the $f_{val}$ parameter, we tune the quantile parameter with the external validation set. Quantiles near the 90th return only modest improvements, and thus we analyze the 90th quantile.

To evaluate whether we overfit to this validation set, we perform k-fold cross validation with the number of folds, $k$, equal to five. We train our models on four out of five folds, tuning the $f_{val}$ parameter such that the model performance agreement is maximized on the testing set. We find that the value of $f_{val}$ that results in superior performance is 0.03, suggesting that our work above generalizes appropriately. The k-fold cross validation results are given in Tab. S2.

We also use the same validation set to compare performance across all four algorithms examined in this study. We evaluate performance of each by calculating the percentage agreement between each algorithm's labels and the validation set labels.

**2.7 Interpretation: k-Means Clustering and PCA**

We perform k-means clustering on the extracted anomalies using the k-means function available in R's base package (R, 2021). We set the number of centers (clusters) to 3 and choose 200 iterations with different random starts to ensure the derived result was robust to utilized starting values. We assign cluster labels based on the cluster means to ensure consistency in label assignment. We use prcomp available in the R base package to calculate principal component loadings



and scores for visualization (R, 2021). We use R packages scattermore (Kratochvil, 2022) and tidyverse (2022) for visualization itself. We perform Varimax rotation using R package psych (Revelle, 2022) to compare to results from a previously published study (Larson et al., 2017).

We create boxplots of assigned roadway trucking variables to probe potential meanings of clustered anomalies. We extract
roadway trucking variables from the Texas Department of Transportation's (TxDOT) roadway inventory (TxDOT, 2022) with processing performed using R package sf (Pebesma et al., 2022). We average records along the same road segment with weights equivalent to the distance between fields in the shapefile FROM_DFO and TO_DFO, which are distance measures representing starting and ending points for those records in the shapefile. Extracted roadway variables from the shapefile include Annual Average Daily Traffic Counts (AADT), Truck AADT Percentage (TRUCK_AADT_PCT), and the number
of all trucks in AADT (AADT_TRUCKS).

**2.8 Census Tract Assignment**

To determine differences in anomaly frequency between census tracts, we assign points (Pebesma et al., 2022) to census tracts using tract boundaries stored in a shapefile used in a previously published analysis of the same campaign data (Miller et al, 2020; Actkinson et al., 2021). We count anomalies of a given cluster assignment and divide by the total recorded
measurements in each polygon. Because each census tract was sampled at different hours from one another and because the objective of the analysis was to compare census tracts, we implement a rescaling procedure described in detail in Sect. S1. As part of that procedure, we restrict the comparisons to 19 of the 35 census tracts, to measurements taken between 8 AM and 4 PM local time, and to measurements taken on weekdays. To account for different polygons containing differing number of measurements, we divide the total amount of rescaled anomaly types by the total number of measurements made
in the census tract, deriving a probability of encountering the specified anomaly type during the campaign in the restricted time interval described above. This probability represents the chance of detection of a given anomaly during the campaign study period. Sect. S2 describes a bootstrapping procedure used to estimate errors associated with these probabilities, which are provided in Tabs. S3, S4, and S5.

**3 Results**

**3.1 External Validation**

We run all four algorithms – Drewnick, QOR, QAND, and DBSCAN –on the Houston mobile monitoring campaign data. To differentiate performance, we compare each algorithm's labeled anomalies with the anomalies of the validation set on the same subset of days, which are considered the ground truth. We observed the algorithm to capture clean conditions as well; the DBSCAN algorithm labeled 848 multivariate realizations with all pollutants lower than their respective 5th quantiles as
noise or just 0.07% of the total number of labeled anomalies.



Of the four algorithms, DBSCAN had the best performance, with its labels exhibiting 86.9% agreement with the validation set's labels. The QOR, QAND, and Drewnick algorithms exhibit 85.5%, 77.0%, and 81.8% agreement, respectively. For context, an algorithm that simply labeled all points as normal would generate 74.7% agreement with the validation set. Because this baseline agreement is so high, we create confusion matrices to probe sources of agreement and disagreement

between each algorithm's predicted anomalies and the validation set labeled anomalies and display them in Fig. 3. Confusion matrices compare how an algorithm categorizes points with the points' true categories. In our work, confusion matrices tabulate the number of points that a given algorithm labels as normal or anomaly that are correspondingly labeled as normal or anomaly in the validation set.

Figure 3 illustrates that even though the DBSCAN algorithm exhibits greater overall agreement with the validation set, it

predicts anomalies less successfully compared to the QOR algorithm. However, the DSBCAN algorithm outperforms the QOR algorithm in its ability to not predict normal points as anomalous. This suggest that the QOR algorithm captures the most anomalies but is a coarse approach to doing so; the DBSCAN algorithm captures fewer anomalies but is less likely to predict something as anomalous when it is not. Table S6 contains counts of instances in which one algorithm made a mistake of a given type when the other did not. Table S6 provides further evidence that the DBSCAN algorithm is inferior in its

ability to label anomalous points compared to the QOR algorithm, while the QOR algorithm is inferior in its ability to not label normal points as anomalous. For the purposes of further analysis, we focus our attention on DBSCAN-derived anomalies, bringing in QOR derived anomalies periodically for comparison. We choose to focus on results from DBSCAN as the approach is more conservative; it does not result in as many false positives as the QOR algorithm and provides confidence that what is being analyzed is an anomaly. The QAND and Drewnick algorithms do not offer superior

performance over the DBSCAN and QOR algorithms, and we do not consider them for further analysis.

### 3.2 k-Means Clustering and PCA

We cluster detected anomalies using R function kmeans, which consistently yields one cluster rich in $CO_2$ concentrations ("$CO_2$ Cluster"), another cluster that contains lower (but still higher than their non-anomaly counterparts) concentrations of all four pollutants for both QOR and DBSCAN derived anomalies ("Transition Cluster"), and a third cluster rich in

BC/$NO_x$/UFP ("BC/UFP Cluster") concentrations. Table 1, Fig. 4, and Fig. S5 contain statistics describing the contents of each cluster. The results are consistent with previously published emissions patterns associated with light and heavy-duty vehicles. Heavy-duty, diesel-powered vehicles emit more BC, $NO_x$, and UFP per kilogram of fuel than light-duty vehicles, often an order of magnitude or more (Dallmann et al., 2012; Dallmann et al., 2013; Park et al., 2011; Preble et al., 2018). Additionally, loadings from the PCA biplot in Fig. S5 when varimax rotated are consistent in split with those reported in

Larson et al. (2017); loadings are sequestered into BC/UFP-rich and $CO_2$-rich factors which are attributed to heavy- and light-duty vehicle activity, respectively. These loadings are given in Tab. S7.

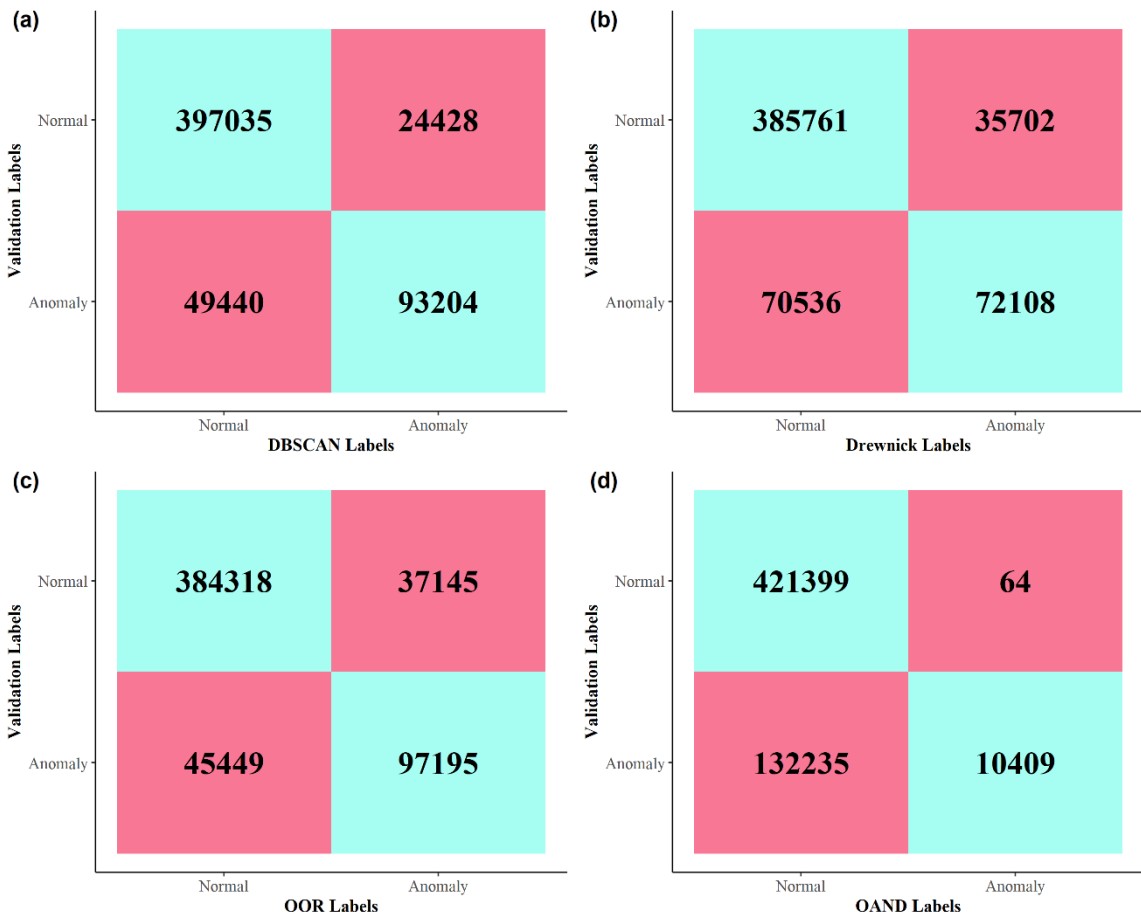

**Figure 3. Confusion matrices corresponding to the performance of (a) DBSCAN, (b) Drewnick, (c) QOR, and (d) QAND. Overall**
**agreement between each algorithm and the validation set was (a) 86.9%, (b) 85.5%, (c) 81.8%, and (d) 77.0%. For example, DBSCAN and the validation efforts both label 397,035 points as normal and 93,204 as anomalous. DBSCAN labels 49,440 points as normal when the validation efforts label them as anomalous; conversely DBCSAN labels 24,428 points as anomalous when the validation efforts label them as normal.**





**Table 1. DBSCAN and QOR k-means cluster means for the four pollutants considered.**

|  | $CO_2$ (ppm) | BC (ng m$^{-3}$) | $NO_x$ (ppb) | UFP (p cc$^{-1}$) |
|---|---|---|---|---|
| **DBSCAN** |  |  |  |  |
| **1$^{st}$ cluster** | 556 | 1893 | 73 | 16298 |
| **2$^{nd}$ cluster** | 444 | 1540 | 43 | 15411 |
| **3$^{rd}$ cluster** | 493 | 6326 | 179 | 50244 |
| **QOR** |  |  |  |  |
| **1$^{st}$ cluster** | 547 | 2142 | 83 | 17463 |
| **2$^{nd}$ cluster** | 444 | 1597 | 42 | 16616 |
| **3$^{rd}$ cluster** | 495 | 6639 | 184 | 51112 |

**Clustered DBSCAN Anomaly Boxplots**

**Figure 4. Boxplots of clustered DBSCAN anomalies by cluster label. Red rectangles correspond to insets of CO₂ and BC that are displayed on the right side of the plot.**






To verify vehicle-related impacts associated with these clusters, we extract traffic variables from the TxDOT roadway inventory and assign these values to our clustered anomalies based on nearest neighbor assignment between the logged GPS coordinates of each clustered point and the latitude/longitude coordinates of the inventory's features (TxDOT, 2022). We plot these assignments in Fig. 5. Panel (a) in Fig. 5 contains the overall AADT counts. Panel (b) in Fig. 5 shows percentages

of trucks in the estimated annual AADT counts. The high percentage of trucks in AADT in the BC/UFP cluster suggests that the cluster is related to trucking activity, while the lower trucking percentage in combination with elevated AADT compared to the transition cluster suggests that the $CO_2$ cluster is capturing light-duty vehicle activity. Results from these boxplots confirm that our clusters are linked to emissions from these different vehicle types.

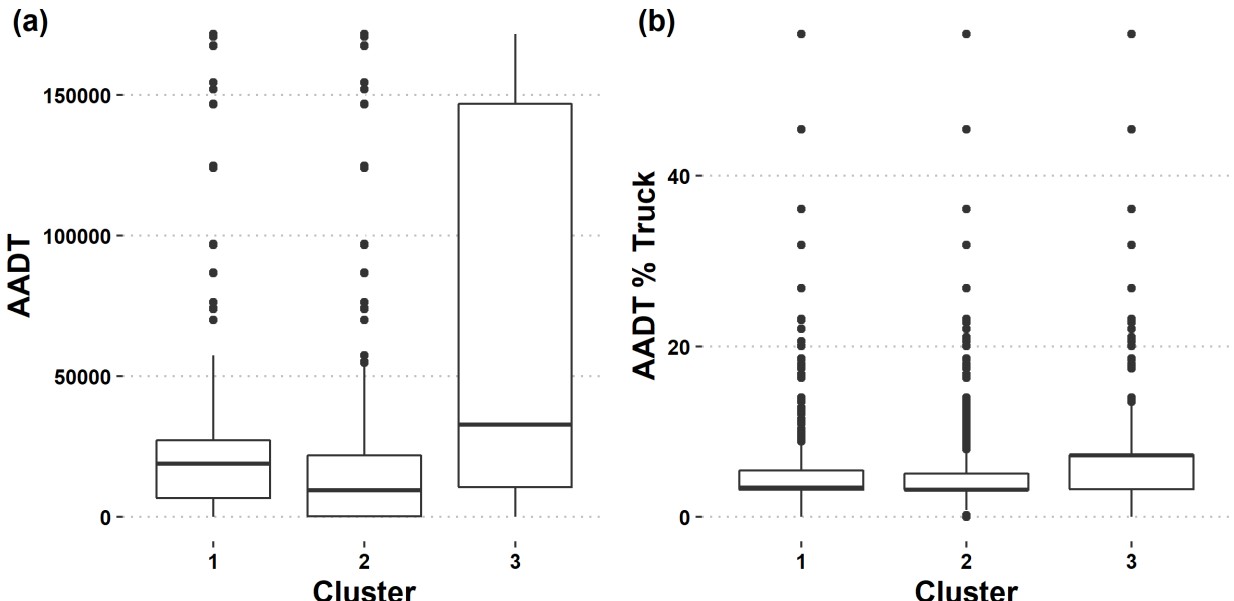

**Figure 5. Boxplot of traffic attributes corresponding to anomalies in labeled clusters. (1-"$CO_2$ Cluster", 2 – "Transition Cluster", 3 – "BC/UFP Cluster"). (a) Annual average daily traffic (AADT) by cluster label. (b) Percentages of trucks in the annual average daily traffic counts (AADT% Truck).**

**3.3 Detected Anomaly Type by Census Tract**

To evaluate spatial differences in these clustered anomaly types across the city of Houston, we tabulate anomaly types for a

subset of visited census tracts; details about the census tracts are provided in Tab. S8. We report rescaled total numbers of detected anomalies of a given cluster type ("$CO_2$ Cluster" for "$CO_2$-rich", "Transition Cluster", "BC/UFP Cluster") divided by the total number of measurements made in that census tract. Normalizing by the total number of measurements in this manner yields the probability of encountering the anomaly in the census tract during the study period, which is from 8 AM to 4 PM local time on weekdays. Figure S6 displays bar plots showing DBSCAN anomaly detection type probabilities by





census tract, while Figs. 6 and 7 map the census tracts colored by their $CO_2$ and BC/UFP anomaly detection type probabilities.

**Figure 6. Map depicting analyzed census tracts colored (darker indicates larger probability) by their calculated $CO_2$ anomaly**
**detection probabilities (%). Wikimedia, 2021. Distributed under the Creative Commons Attribution-ShareAlike 4.0 license. https://foundation.wikimedia.org/w/index.php?title=Maps_Terms_of_Use#Where_does_the_map_data_come_from.3F.**






**Figure 7. Map depicting analyzed census tracts colored (darker indicates larger probability) by their calculated BC/UFP anomaly**
**detection probabilities (%). Wikimedia, 2021. Distributed under the Creative Commons Attribution-ShareAlike 4.0 license. https://foundation.wikimedia.org/w/index.php?title=Maps_Terms_of_Use#Where_does_the_map_data_come_from.3F.**

The bar plots and maps illustrate stark spatial heterogeneity in anomaly type. With respect to $CO_2$ cluster anomalies, neighborhoods in the western parts of Houston (North Rice, South Rice, Sharpstown) consistently rank higher than
neighborhoods in the eastern part of Houston (Milby Park, Clinton, Manchester), with neighborhoods surrounding Rice University ranking the highest. The neighborhoods near the Rice campus consist of busy thoroughfares that are often congested with traffic from light-duty gasoline powered vehicles, especially around local rush hour (8 AM). With regards to the BC/UFP clusters, heavily industrialized neighborhoods in the eastern part of Houston near the Houston Ship Channel (Milby Park, West Galena Park, Manchester, Clinton) are ranked the highest, with the Milby Park census tract exhibiting the
highest probability of encountering one of these anomaly types (10.6%) during the study period.

Many of the BC/UFP anomaly detections occur on highway; Figure S7 illustrates the differences in BC/UFP anomaly detection probabilities when highways are included and excluded from the analysis (Figure S8 shows the same information



for $CO_2$ anomalies). Even with highways removed from the analysis, neighborhoods in the eastern part of Houston still rank consistently higher than those neighborhoods in the western part of Houston with respect to the frequency of BC/UFP

anomaly detection. The mapped census tracts show spatial discrepancies between $CO_2$ dominated and BC/UFP dominated areas with respect to probability of anomaly type detection. Table 2 details probabilities of detecting each anomaly type by census tract, underscoring these spatial disparities. For example, the bold, italicized entries in Tab. 2 indicate a $\approx$ 10x greater chance of encountering a BC/UFP anomaly type in the Manchester census tract compared to the North Rice census tract. These disparities, and the presented evidence suggesting that the BC/UFP anomalies are closely related to heavy-duty

vehicles, are consistent with previous modeling studies that show large contributions of heavy-duty vehicles to air pollution in Houston's Ship Channel (HSC) neighborhoods and previous work pointing out elevated heavy-duty vehicle activity in the HSC area (Zhang et al., 2017; Demetillo et al., 2020).

**Table 2.** Tabulated anomaly detection probability type ("$CO_2$ – rich" = "$CO_2$ %", "Transition" = "Transition %", "BC/UFP – 330 rich" = "BC/UFP %") by census tract.

| Census Tract | $CO_2$ % | Transition % | BC/UFP % | Total Collected Observations |
|---|---|---|---|---|
| Bayland Park | 1.7 | 8.6 | 0.8 | 138367 |
| Washington Corridor | 2.8 | 13.3 | 1.9 | 206611 |
| Manchester | 0.8 | 19.6 | *5.6* | 97374 |
| East Galena Park | 0.7 | 8.6 | 0.7 | 77046 |
| Milby Park | 1.2 | 16.8 | 10.6 | 110019 |
| Sharpstown | 4.6 | 17.8 | 2.8 | 80560 |
| Sharpstown South | 2.2 | 9.5 | 1.3 | 114595 |
| West Galena Park | 1.5 | 16.5 | 6.0 | 134501 |
| North Spring Branch | 2.1 | 12.0 | 1.0 | 100391 |
| North Rice | 5.8 | 14.4 | *0.6* | 263585 |
| Clinton | 1.2 | 20.1 | 4.4 | 185196 |
| West Eastex | 1.1 | 12.8 | 2.5 | 144963 |
| North Heights | 1.4 | 10.4 | 1.4 | 246103 |
| South Rice | 5.0 | 13.4 | 0.6 | 139313 |
| Harrisburg | 1.0 | 16.9 | 4.2 | 127736 |
| Sharpstown North | 3.6 | 18.7 | 1.2 | 98743 |
| Westchase | 3.4 | 12.7 | 1.3 | 68620 |
| South Spring Branch | 2.3 | 13.3 | 2.4 | 78195 |
| South Beltway Central | 0.9 | 16.3 | 2.2 | 311589 |





## 4 Conclusions

We discuss the successful development of a new approach to detect plumes in mobile monitoring time series using an anomaly detection algorithm based on DBSCAN and use the resulting analysis to derive anomaly frequencies representative

of different emission impacts in different Houston neighborhoods. While previous work has implemented DBSCAN in conjunction with deep learning models to analyze satellite $PM_{2.5}$ measurements (Lu et al., 2021) or used it to define microenvironments in air pollution exposure contexts (e.g., home, work, or restaurant) (Do et al., 2021), this is the first study to incorporate DBSCAN in plume detection efforts. The algorithm offers comparable, if not superior, performance to previously published plume detection techniques for mobile monitoring time series and is justified in analyses warranting a

conservative approach. In this work, we show how this approach illustrates different emission impacts in census tracts around the city of Houston. Specifically, we show how BC/UFP anomaly frequencies were ≈ 10x greater in census tracts in the eastern part of Houston near the HSC compared to neighborhoods in the western part of Houston. While it is not definitive that this cluster type represents impacts from heavy-duty vehicles, for there is no observational evidence to connect those observations to those vehicle types directly, anomaly emission patterns are consistent with previously

published studies analyzing emissions from light and heavy-duty vehicles (e.g., Larson et al. (2017) and references therein). Previous studies also have shown the large impacts of trucking on pollution in the HSC area and have raised environmental justice concerns with the burden of pollution from diesel-powered vehicle activity (Demetillo et al., 2020; Zhang et al., 2017). Results from this work emphasize the need for additional investigation into the trucking activity in HSC neighborhoods, and, more broadly, illustrate how mapped spatial distributions of these anomalies can be used to inform

regulatory activities.

Results from this algorithm could be incorporated into health assessment frameworks. Clustered anomalies could be grouped into source categories to facilitate simple exposure estimates from different sources. Apportioning anomalies to nearby sources and determining their frequencies would be an interesting approach to determining whether some sources are more harmful to health than other sources. Census-tract weighted probabilities of an anomaly could be employed in random walk

simulations of cumulative air pollution exposure, providing a different metric to evaluate related health effects (Tang and Niemeier, 2021). Future work could focus on addressing serial dependency inherent in detected anomalies to develop probability-based exposure estimates, as well as the general development of a framework that relates health outcomes to the frequencies of these detected anomalies.

There are opportunities to improve this algorithm in future work. This algorithm should be evaluated using different external

validation methods, such as having an observer sit in the vehicle and noting emissions events while the vehicle is measuring to create the validation set. Alternative nearest neighbor clustering techniques could be explored; local outlier factors could be used to address situations where DBSCAN does not exhibit great performance (Tan et al., 2019). Finally, an ensemble



approach utilizing both DBSCAN and other clustering techniques could be investigated for improved performance (Drewnick et al., 2012; Actkinson et al., 2021).


## Code Availability

A GitHub repository containing code used to generate the work is available here: https://zenodo.org/badge/latestdoi/449031959

Additionally, an R Shiny application containing a graphical user interface to the software is available at the following URL: https://bactkinson.shinyapps.io/plume_detection_with_dbscan/. The doi for the repository containing code used to generate the Shiny app is available here: https://zenodo.org/badge/latestdoi/483829076

## Data Availability

Validation datasets used in this work are available at the following Zenodo repository doi: 10.5281/zenodo.6473859

## Author Contributions

BA conceived, wrote, and analyzed the plume detection algorithm with helpful insight from RG. BA wrote the manuscript. RG provided helpful edits and suggestions.


## Competing Interest Statement

The authors declare that they have no conflict of interest.

## Acknowledgements

The following R packages were used in the analysis and visualization of results: tidyverse (2022), ggpubr (Kassambara, 2020), caret (Kuhn), dbscan (Hahsler et al., 2019), leaflet (2022), leafem (Appelhans et al., 2021), sf (Pebesma et al., 2022), mapview (2022), scattermore (Kratochvil, 2022), base (R, 2022), and data.table (Dowle et al., 2021). The authors gratefully acknowledge the support of NIEHS (grant #R01ES028819-01). Additionally, we appreciate the support of Environmental Defense Fund for the collection and provision of the mobile data used to develop this algorithm. Finally, we acknowledge
Dr. Katherine Ensor and Dr. Daniel Cohan for useful suggestions and input.

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
