# Peer review of "Detecting plumes in mobile air quality monitoring time series with Density-based Spatial Clustering of Applications with Noise"

_Atmospheric Measurement Techniques, 2023_

## Referee Comment (RC1)

**Detecting plumes in mobile air quality monitoring time series with Density-based Spatial Clustering of Applications with Noise**
*Atmos. Meas. Tech.*
Actkinson and Griffin

**General Comments:**
Overall, this article presents a novel method for identifying and isolating plumes from mobile timeseries measurements. The proposed technique produces a conservative grouping of plumes which can significantly progress analyses and use of mobile data for local air quality and inequality analyses. The application of this technique on a wider scale would be particularly interesting for further addressing differences in air pollution burden between communities. This paper presents useful technique especially with the increase in mobile measurements, but further discussion on source apportionment after plume identification is encouraged.

**Specific Comments:**
The methodology is sound but discussion of pollutant co-emissions from sources may need to be considered when identifying sources through this method. For instance, the authors site the sources between heavy- and light-duty vehicles, and while the Houston shipping channel is a traffic pollution hotspot, a large portion of the sources are stationary point sources such as petrochemical and industrial facilities. This is particularly important when extending the results of this analysis to census tracts where multiple pollutant sources dominate. Additionally, conducting an additional validation using mobile data nearfield of ground-based monitors may bolster plume identification.

**Technical Corrections:**
*n/a*

---

## Author Response (AR1)

REVIEWER 1 COMMENTS

The methodology is sound but discussion of pollutant co-emissions from sources may need to be considered when identifying sources through this method. For instance, the authors site the sources between heavy- and light-duty vehicles, and while the Houston shipping channel is a traffic pollution hotspot, a large portion of the sources are stationary point sources such as petrochemical and industrial facilities. This is particularly important when extending the results of this analysis to census tracts where multiple pollutant sources dominate. Additionally, conducting an additional validation using mobile data nearfield of ground-based monitors may bolster plume identification.

AUTHOR RESPONSE

We thank this referee for taking the time to provide their feedback. We agree that careful consideration of sources is important in interpreting the results of this method. Our previous work (Miller et al., already cited) discusses the impact of elevated point sources on this data.  We believe that using stationary monitors that include a wide suite of measurements would prove invaluable to provide accurate source identification but is beyond the scope of this study, as the mobile platform did not co-locate with any stationary monitor for an extended period of time. We have added the following lines to the conclusion to incorporate this suggestion as a direction for future research.

(at ~L357) "There are opportunities to improve this algorithm in future work. For example, this algorithm should be evaluated using different external validation methods, such as having an observer sit in the vehicle and note emissions events (for example, driving behind a heavy-duty diesel vehicle) while data are being collected to create the validation set. Additionally, the mobile platform could be co-located with a wide suite of stationary instruments to enable more confidence in source identification."

REVIEWER 2 COMMENTS

1. The need to separate emissions from other vehicles on the roadway from "ambient" conditions is a persistent challenge in interpreting mobile measurement data sets.  This proposed technique improves on current methods and should allow for better analysis and interpretation of the data.  Many of the areas investigated were bounded by major roadways or sources yet only one value was given for the entire tract.  While tract level information can be useful where the tract is relatively homogenous, it can also misrepresent the degree of impact in heterogenous tracts.  I would be curious to see whether the results would be similar if a smaller grid were used rather than census tracts. Overall, the paper is well written and I anticipate it will aid the community in analyzing mobile measurement data sets.

2. Line 239-241 I found the sentence about Table S6 confusing and had to re-read it several times.  Perhaps this could be rewritten more clearly.

3. Line 316-318  If there is room, I would suggest including this figure in the main paper as I feel it is important to help characterize the impact of major roadways.

AUTHOR RESPONSE

1. We thank this referee for taking the time to provide their feedback. We acknowledge the pitfalls that can occur in assigning a single value to an entire census tract when there is significant variation within the census tract. With this method, it is possible to go to finer distance scales subject to instrument and sampling constraints. Future work could explore the tradeoffs made in increasing spatial coverage at the expense of decreasing temporal coverage and the effects of this on the resulting interpretation.  It should also be noted that demographic data often are available only at the census-tract level.  One sentence was added at the end of the response to reviewer 1:

"Future work also could consider aggregating data on a scale finer than a census tract to address heterogeneity of emissions within a census tract."

2. We have written the caption for Table S6 to be the following to address the reviewer request:

"Specific label counts in which the QOR algorithm underperforms or overperforms relative to the DBSCAN algorithm."

3. We have moved the figure referenced by the reviewer into the main paper.